# Advancements and Insights in Exosome-Based Therapies for Wound Healing: A Comprehensive Systematic Review (2018–June 2023)

**DOI:** 10.3390/biomedicines11082099

**Published:** 2023-07-25

**Authors:** Patrícia Sousa, Bruna Lopes, Ana Catarina Sousa, Alícia Moreira, André Coelho, Rui Alvites, Nuno Alves, Stefano Geuna, Ana Colette Maurício

**Affiliations:** 1Departamento de Clínicas Veterinárias, Instituto de Ciências Biomédicas de Abel Salazar (ICBAS), Universidade do Porto (UP), Rua de Jorge Viterbo Ferreira, No. 228, 4050-313 Porto, Portugal; pfrfs_10@hotmail.com (P.S.); brunisabel95@gmail.com (B.L.); anacatarinasoaressousa@hotmail.com (A.C.S.); alicia.moreira.1998@gmail.com (A.M.); andrefmc17@gmail.com (A.C.); ruialvites@hotmail.com (R.A.); 2Centro de Estudos de Ciência Animal (CECA), Instituto de Ciências, Tecnologias e Agroambiente da Universidade do Porto (ICETA), Rua D. Manuel II, Apartado 55142, 4051-401 Porto, Portugal; 3Associate Laboratory for Animal and Veterinary Science (AL4AnimalS), 1300-477 Lisboa, Portugal; 4Instituto Universitário de Ciências da Saúde (CESPU), Avenida Central de Gandra 1317, 4585-116 Paredes, Portugal; 5Centre for Rapid and Sustainable Product Development, Polytechnic of Leiria, 2430-028 Marinha Grande, Portugal; nuno.alves@ipleiria.pt; 6Department of Clinical and Biological Sciences, Cavalieri Ottolenghi Neuroscience Institute, University of Turin, Ospedale San Luigi, 10043 Turin, Italy; stefano.geuna@unito.it

**Keywords:** animal models, exosomes, skin regeneration, wound healing, systematic review

## Abstract

Exosomes have shown promising potential as a therapeutic approach for wound healing. Nevertheless, the translation from experimental studies to commercially available treatments is still lacking. To assess the current state of research in this field, a systematic review was performed involving studies conducted and published over the past five years. A PubMed search was performed for English-language, full-text available papers published from 2018 to June 2023, focusing on exosomes derived from mammalian sources and their application in wound healing, particularly those involving in vivo assays. Out of 531 results, 148 papers were selected for analysis. The findings revealed that exosome-based treatments improve wound healing by increasing angiogenesis, reepithelization, collagen deposition, and decreasing scar formation. Furthermore, there was significant variability in terms of cell sources and types, biomaterials, and administration routes under investigation, indicating the need for further research in this field. Additionally, a comparative examination encompassing diverse cellular origins, types, administration pathways, or biomaterials is imperative. Furthermore, the predominance of rodent-based animal models raises concerns, as there have been limited advancements towards more complex in vivo models and scale-up assays. These constraints underscore the substantial efforts that remain necessary before attaining commercially viable and extensively applicable therapeutic approaches using exosomes.

## 1. Introduction

The skin serves as the body’s external protection against harmful agents, regulating the internal temperature and integrity while maintaining homeostasis. Under normal conditions, the skin can regenerate itself through a complex process that comprises four distinct phases: hemostasis, inflammation, proliferation, and remodeling. However, when this process fails or is disrupted, it can culminate in impaired tissue regeneration or prolonged wound healing, leading to the formation of chronic wounds [1,2,3].

Chronic wounds are characterized by a prolonged inflammation (lasting from 4 to 12 weeks), often associated with infections, microbial biofilms, and impaired response from epithelial cells [4]. These wounds are multifactorial and frequently occur in individuals with several diseases, including diabetes, infections, and arterial/venous insufficiency [5].

The prevalence of chronic non-healing wounds is increasing due to factors such as population aging and aging-associated diseases, concomitant diseases, tumors, and congenital defects, negatively impacting the quality of life of millions of people worldwide. Therefore, the socioeconomical and health care burden is also increasing [1,2,3].

Current therapies such as debridement, antibiotherapy, and dressings remain insufficient, as they are not efficient, and there is still a need for new treatments. In the last few years, regenerative medicine has gained popularity, and extensive research has been conducted on mesenchymal stem cells (MSCs) and their derivates in several fields, including wound healing and skin regeneration.

MSCs are defined by the International Society of Cellular Therapy as similar to fibroblasts, adherent to plastic, and with the ability to differentiate into three different cell lines in vitro (chondrocytes, osteoblasts, and adipocytes). These cells should express the surface markers CD73, CD90, and CD105 while not expressing hematopoietic markers (CD14, CD45, CD34, CD19/HLA-DR, and CD11b/CD79). These undifferentiated cells have the potential to repair different tissues as they undergo differentiation. Furthermore, MSCs can be obtained from different sources and species. In recent years, there has been a notable rise in the use of MSCs for wound-healing purposes, with numerous studies showcasing promising outcomes utilizing cells obtained from diverse sources [1,2]. 

The use of MSC-derived products, such as secretome and exosomes, when compared to MSCs offers some advantages, including a reduced risk of tumorigenesis and minimal immune rejection [4], which is steadily rising. Exosomes are nanovesicles secreted from the endosomal system (30–150 nm) and represent one of the three major subpopulations of extracellular vesicles [6]. The other subpopulations include apoptotic bodies (>100 nm) and microvesicles (100–1000 nm). Exosomes are produced by several different cell types from different origins [7]. 

Previous research has highlighted the crucial role of exosomes in facilitating cell-to-cell communication, namely by sharing their cargo as miRNAs and proteins. Several studies have indicated that exosomes obtained from stem cells have the potential to assist with and promote tissue repair. This is attributed to their unique advantages, including exceptional stability, minimal risk of immune rejection, targeted delivery to specific tissues, straightforward control of dosage, and definable concentration [8].

Several studies have shown that exosomes derived from MSCs have similar therapeutic properties, angiogenic ability, and immune modulation as the cells from which they originate [3,9,10,11].

The quality of wound healing relies on the fibroblast’s migration and proliferation as well as collagen synthesis and deposition. Exosomes, with their abundant content of RNAs and proteins relevant to fibroblast functions, are thought to facilitate these processes. This optimization of fibroblast activities ultimately contributes to the accelerated wound-healing mechanism [8].

Multiple studies have demonstrated the therapeutic potential of exosomes in various stages of wound healing. In the inflammation phase, exosomes have been shown to modulate immune cells and resident tissue cells, leading to a reduction in uncontrolled inflammatory responses. During the proliferation phase, exosomes play a role in wound closure by activating endothelial cells and fibroblasts. This activation promotes a proangiogenic environment and initiates the deposition of extracellular matrix. In the remodeling phase, exosomes influence the balance between matrix metalloproteinases and tissue inhibitors of matrix metalloproteinases, favoring optimal wound-healing outcomes. Exosome therapy also enhances wound healing by stabilizing and stimulating a wide range of mediators involved in each phase [10].

The aim of this systematic review was to access the exosome-based therapies wound-healing effects and methodological heterogeneity in studies and furnish the scientific community with a comprehensive overview of the advancements made in this field over the past five years. Additionally, the review encompasses an analysis of the animal models used to evaluate the translational potential of the data to human medicine. 

## 2. Data and Methods

This systematic review was performed according to PRISMA (Preferred Reporting Items for Systematic Reviews and Meta-Analyses) guidelines.

The main goal of this study was to assess the methodology heterogeneity and provide insights into the progress of exosome research in wound healing over the past five years. 

The research involved using the PubMed database, covering the period from 2018 to June 2023. The search query used the following keywords: “wound healing” (Title/Abstract) AND “exosomes” (Title/Abstract) NOT “review” (Title/Abstract), which initially retrieved 531 results.

All 531 publications underwent title, abstract, and full-text article examination.

The eligibility criteria included: (1) English language, (2) full access to the publication, (3) use of exosomes from mammalian sources, (4) exosomes applications in wound healing, and (5) use of animal models (in vivo studies).

Exclusion criteria were applied to filter out the studies that did not meet the research goals, such as: (1) in vitro studies, (2) review articles, (3) non-English-language publication, (4) studies unrelated to wound healing, (5) no full access to the publication, and (6) studies not involving exosomes.

To ensure rigorous study selection, all authors participated in the process and conducted double checks. Any discrepancies or disagreements were resolved through discussion and consensus. Duplicates were searched through Endnote software, and data analysis was performed using an Excel form specifically designed by the authors.

Among the initial 531 studies, 383 were excluded based on the exclusion criteria mentioned above, resulting in 148 papers. These remaining papers underwent a thorough double check by all authors.

The extracted information from the selected studies included PMID (PubMed Identification), paper title, publication date, corresponding author’s country, cell species, cell type, biomaterial usage, administration route, animal models, and exclusion criteria (when applicable).

GraphPad Prism version 8.0.1 was used to elaborate the graphical representations of the collected data.

Bias assessment was evaluated for each study regarding the adherence to MSCs minimal criteria, and animal models were examined to determine external validity.

The selection process is summarized in Figure 1. 

Through this systematic review, the study aimed to provide valuable insights and contribute to the understanding of exosome-derived treatments in wound healing, with potential implications for potential translation into human medicine.

## 3. Results

### 3.1. Retrieved Data

The following table summarizes the retrieved data from the selected 148 papers (Table 1).

### 3.2. Scientific Data Production and Publication Distribution between 2018 and June 2023

All 148 papers selected were comprehensively analyzed to assess the temporal distribution of their publication across the five-year timeframe (Figure 2). In 2018, a total of seven papers were published (4.7%). The subsequent year, 2019, witnessed a notable increase in publications, with 16 papers, accounting for 10.8% of the selection. The publication rate continued to rise in 2020, reaching 30 papers (20.3%). In 2021, 29 papers were published (19.6%). The most significant publication rate occurred in 2022, with 42 papers published (28.4%). Up until June 2023, 24 additional papers had already been published, indicating that the year will probably surpass previous records (16.2%). These data demonstrate that there has been a continuous and stable increase in scientific investment in this research area, with the number of works carried out and published results increasing continuously.

The geographical distribution of scientific publications in the field of exosome application in wound healing was examined, focusing on the corresponding author’s country over a 5-year timeframe (Figure 3). China emerged as the country with the highest scientific publication rate in this field, with 122 publications (82.4%). The United States of America (USA) followed with eight publications (5.4%), South Korea with five publications (3.4%) and Portugal with three publications (2.0%). Other countries have lower publication rates: Japan recorded two publications (1.4%), as did Iran (1.4%) and India (1.4%). Similarly, Egypt and Taiwan each had one publication (0.7%). Additionally, a collaboration between China and Finland resulted in two publications (1.4%). 

### 3.3. Cell Source and Type

Cell source was analyzed as seen in Figure 4. Among the papers analyzed, human tissues were the preferred cell source for exosome extraction (73.6%). Then, rodent-derived cells corresponded to 23%, with 15.5% from mice and 7.4% from rat. The other sources, although less frequent, consisted of dog (0.7%) and macaque (0.7%). Furthermore, one study compared mice- and rat-derived exosomes (0.7%), while two independent studies compared both mice- and human-derived exosomes (1.4%). 

According to the data presented in Table 2, the most commonly used to extract exosomes in humans is the adipose tissue (ADSC) with 26.6%, followed by the UCMSCs (umbilical cord mesenchymal stem cells) with 22.9%, and the BMSCs (bone-marrow-derived mesenchymal stem cells) with 10.1%. Other tissues still present considerable percentages of use, such as placenta (2.8%), peripheral blood (3.7%), epidermal (3.7%), DP (dental-pulp-derived mesenchymal stem cells) (2.8%), and umbilical vein (7.3%). Regarding mice, adipose tissue emerges as the preferred source (30.4%), followed by bone marrow (26.1%). In rats, both adipose tissue and bone marrow (36.4%) are favored as primary sources for exosome extraction. These results are in agreement with the general scientific literature related to the use of cell-based therapies, where adipose tissue, bone marrow, and umbilical cord are the most explored tissues, and the respective MSCs are the most studied and characterized cells both for their direct use and use of their secretion products.

### 3.4. Biomaterials and Administration Route

As seen in Figure 5, the analysis of biomaterials used in all 148 papers revealed that the majority of studies chose to use exosomes without any biomaterial (73.6%). However, when a biomaterial was selected, hydrogels were the most commonly used (18.2%). Other biomaterials were also employed in several studies, such as scaffolds (2.7%), patches (1.4%), sponges (2.0%), nanoparticles (0.7%), and dressings (1.4%).

In the analysis of these 148 papers, the administration route of exosomes was analyzed and compared to the selected biomaterials, as illustrated in Figure 6. The preferred method, regardless of the presence of biomaterials, was via subcutaneous (SC) injection at the wound margins (66.2%). Within this route, subcutaneous injection without a biomaterial (57.8%) was the most common approach, while the association with a biomaterial was only 8.8%.

Topical administration was used 25.7%, with 16.9% involving the use of a biomaterial and 8.8% without one. Other administration routes included intradermal (ID) injection (2.7%), intraperitoneal (IP) injection (1.4%), and endovenous (EV) injection (2.0%). There were also two papers that compared topical and subcutaneous injections (1.4%), while one study compared topical and intradermal injections (0.7%).

### 3.5. Animal Models

Among the studies included in the analysis, rodents were used in the majority, accounting for 96.6% of the total (143 studies), with mice comprising 66.9% (99 studies) and rats 29.7% (44 studies). Also, two studies used both mice and rats (1.4%), while one study used a non-human primate model (0.7%), and the last used a canine model (0.7%). Figure 7 provides a visual representation of the distribution of in vivo models used in the selected studies.

## 4. Discussion

### 4.1. Scientific Data Production and Publication Distribution between 2018 and June 2023

The publication rate regarding the use of exosomes for wound healing has shown a significant increase in the last five years, with 2022 marking the highest publication rate to date. Based on the count of 24 publications as of June, 2023 is expected to surpass previous records. 

This notable increase in scientific publications reflects the recent emergence and promising outcomes of exosome-based therapies in wound healing. It is expected that even more valuable data will be published in the next years.

In addition, China stands out as the leading country regarding publication rate, which demonstrates the high importance this topic in this country. However, there remains a gap in research development and publication in this field in other countries, particularly in Europe and America. It is essential to encourage research and publication in these regions in order to promote advancements in the field of exosome-based therapy worldwide. 

### 4.2. Cell Source and Type

Among the preferred tissue source for exosome production, human-derived cells accounted for 73.6% of the studies, followed by rodent-derived cells at 23%, with 15.5% from mice and 7.4% from rat. 

Within the human-derived cells, the most commonly used tissue is the adipose tissue (26.6%), followed by the umbilical cord (22.9%) and the bone marrow (10.1%). In mice, adipose tissue emerges as the preferred source (30.4%), followed by bone marrow (26.1%). In rats, both adipose tissue and bone marrow (36.4%) are favored as primary sources for exosome extraction.

Given that the ultimate objective of most studies is the development of exosome-based therapies for human medicine, the retrieval of exosomes from human tissues appears to be a logical approach. However, the significant heterogeneity within the tissues poses challenges in comparing results, as researchers have not reached an understanding of the most efficacious treatment option.

The preferred use of ADSCs (adipose-tissue-derived mesenchymal stem cells) is probably due to its low ethical issues, easy extraction, and cost-effectiveness. ADSCs have shown potential in wound healing by increasing vascularization, fibroblasts migration, and differentiation and upregulating macrophages chemotaxis [2,160,161].

BMSCs have also demonstrated great potential in wound healing, increasing angiogenesis, and reducing wound contraction [161]. Additionally, BM-MSCs have garnered significant attention as the most extensively investigated subset of MSCs and have been recognized for their relatively low immunogenicity [2].

UCMSCs have also shown their wound-healing potential, as they can differentiate into epidermal tissue and are easier to harvest than BM-MSCs [1,2].

The variation observed in exosomes derived from different species and tissues still needs further research and understanding. The goal should be to identify the most effective sources and optimize and standardize the isolation processes to ensure consistent and reliable outcomes. By doing so, the scientific community can more accurately make comparisons between studies and advance towards the development of efficacious therapeutic approaches. 

In addition, it is valuable to explore alternative sources of exosomes beyond those currently described, as it may uncover potential benefits and characteristics, broadening the range of therapeutic options.

Overall, while human-derived exosomes remain the preferred choice due to their ultimate clinical relevance, efforts should be made to refine the current methodologies and promote collaborations between research groups to better understand the most effective exosome-based treatments. 

### 4.3. Biomaterials and Administration Route

In most studies, exosomes were administered via SC injection in the wound margins, either without a biomaterial (57.8%) or in combination with a biomaterial (8.8%). This delivery method offers ease, speed, and localized treatment administration. Considering that most skin wounds are created on the animal’s dorsum, incorporating biomaterials can be difficult, requiring prior development and testing. SC injection of MSCs has also demonstrated great results regarding wound closure, angiogenesis, and re-epithelization. Alternatively, topical administration is also used (25.7%), as it is less invasive and less painful than the injection methods [162].

However, when a biomaterial was selected, it was predominantly a hydrogel (18.2%). Other biomaterials were also applied in several studies, including scaffolds (2.7%), patches (1.4%), sponges (2.0%), nanoparticles (0.7%), and dressings (1.4%).

The combination of biomaterials aims to improve the therapeutic functionality of exosomes by stabilizing them and prolonging their release at the wound site, thereby preventing rapid entry into blood circulation and systemic dilution. Hydrogels, specifically, offer several advantages in wound healing, such as antibacterial activity, facilitation of tissue adhesion, protection against UV radiation, hemostatic capacity, promotion of spontaneous regeneration, and easy injectability. They can also provide a 3D environment and mimic the extracellular matrix while maintaining proper moister levels at the wound site. Therefore, the use of exosomes associated with hydrogels has shown to improve wound healing, enhancing re-epithelization and vascularization [4,163].

Hydrogels based on chitosan or methylcellulose are considered great options for diabetic wound treatment and have been used in some of the selected studies. These polymers have good biodegradability and biocompatibility and are nontoxic. Geng et al. developed a loaded carboxyethyl chitosan hydrogel loaded with bone-marrow-derived exosomes to improve chronic diabetic wound healing. It increased angiogenesis and neovascularization, reduced local inflammation, and improved wound healing in diabetic rats [27].

Pluronic F-12 hydrogels have also been used in selected studies, as they are injectable, biocompatible, and thermosensitive. Zhou et al. used a pluronic F-12 hydrogel combined with adipose-tissue-derived exosomes to improve re-epithelization, angiogenesis, collagen synthesis, and wound healing and cellular proliferation in mice [17]. Yang et al. also used a similar hydrogel combined with human umbilical-cord-derived exosomes with increased wound-closure rate and granulation tissue in rats [13].

Gelatin methacryloyl (GelMA) hydrogels were also chosen in different studies due to their mechanical properties and ability to retain exosomes for a prolonged time. Zhao et al. used human umbilical vein endothelial cells (HUVECs)-derived exosomes in association with a GelMA hydrogel and demonstrated an improvement of angiogenesis and collagen maturity in rats [56]. Hu et al. used a similar hydrogel with ADSCs-derived exosomes and showed an improvement of wound healing with increased blood vessel regeneration, proliferation, and migration in mice [43].

Several studies have combined hydrogels and MSCs with promising results in skin regeneration. The use of BMSCs seeded into hydrogels improved angiogenesis and accelerated wound healing in mice [164]. Another study demonstrated reduced scar formation and improved angiogenesis, collagen, granulation, and re-epithelization in rabbits through ADSCs combined with a hydrogel [165].

These findings emphasize the importance of carefully selecting the biomaterials and administration route to optimize the therapeutic effects of exosomes in wound healing. 

### 4.4. Animal Models

The results showed that rodents are the main animal models in studies involving exosome-based therapies in wound healing (96.6%). These findings are consistent with Al-Masawa et al.’s previous findings up until March 2021 [3].

The use of small-animal models has several advantages, such as researchers’ familiarity, easy handling, affordability, and availability. However, there are also limitations associated with these models, including skin thickness, fast hair growth cycles, follicular pattern, and wound size [166]. Rodents exhibit a thin epidermis and loose skin adherence along with dense hair that has been suggested to potentially enhance the wound-healing rate. In addition, these animals lack apocrine and eccrine glands but possess a subcutaneous panniculus carnosus muscle that enhances rapid wound contraction. Moreover, they also have stronger immune systems and have endogenous sources of vitamin C, which plays a significant role in wound healing [167,168].

Although rodent models are frequently used in the initial stages of new therapy approaches, it is necessary to scale up to more complex animal models to better reflect the similarities between such models and the human species. The main goal of most researchers is to develop new treatments options for non-healing chronic wounds and make them commercially available to the human population. Therefore, the consistent use of rodents in 96.6% of studies over the last five years limits the broader application of these data.

Although the use of exosome-based therapies has been showing promising results over last few years, the inclusion of larger animal models, such as ovine, swine, dog, and non-human primates, is crucial. In addition, it is important to fulfill the 3Rs principle (replace, reduce, and refine) regarding animal use, implying that research data should evolve until commercialization becomes possible [169,170].

However, these more complex models present challenges, as they are more expensive, more difficult to handle, and require large set-ups. Pigs, for instance, are regarded as standard models for wound-healing research due to the resemblance of their skin to that of humans. They also present physiological and anatomical similarity to the human species. Nonetheless, to date, no studies have been conducted on this particular species. Non-human primates, although sharing greater similarity with humans, are rarely used mainly due to ethical concerns [166,167]. 

The porcine model has been used in wound-healing research with promising results. In particular, the administration of BMSC and ADSC intradermally into partial-thickness wounds enhanced local epithelization and improved wound appearance when compared to the control [171,172]. This suggests that the use of these cells can accelerate the wound-healing process.

Martinello et al. used the ovine model in wound healing and achieved great results. The local injection of peripheral-blood MSCs into the wound margins revealed improved re-epithelization, proliferation, neovascularization, and contraction, with a higher wound-closer rate [173].

In dogs, the use of MSCs to treat chronic wounds has also demonstrated great potential. UCMSCs used in association with a PVA hydrogel showed significant progress in wound regeneration and decreased local ulceration [174]. Other study using ADSc also improved re-epithelization, reduced local inflammation, and promoted epidermal and dermal regeneration in both acute and chronic wounds [175]. In the selected study using a dog model, Bahr et al. used BM-MSCs-derived exosomes in association with a carboxymethylcellulose hydrogel. The results were promising, as the treatment enhanced wound healing with no scaring, with organized collagen deposition and increased dermal fibroblasts [158].

Lu et al. used autologous and allogeneic iPSCs-derived exosomes to improve wound healing in macaques. It demonstrated an increased angiogenesis, collagen deposition, epithelial coverage, and wound-closure rate [56].

While acknowledging the differences among animal models, current approaches in wound healing remain highly relevant. However, it is crucial to increase the use of diverse and more complex animal models to bridge the gap between current findings and their practical application in the human species [176]. 

All the selected papers consistently reported positive outcomes in vivo, highlighting the ability of exosomes to promote wound healing. 

Exosome treatment was found to enhance wound-closure rates, stimulate local angiogenesis and reepithelization, and facilitate collagen deposition [12,14,17,18]. Furthermore, exosomes promoted a reduction in scar formation and decreased local inflammation in multiple studies [18,23,158]. Additionally, exosome treatments resulted in increased granulation tissue formation and enhanced the proliferation and migration of dermal fibroblasts [65,155,157].

The findings consistently indicate that exosomes possess therapeutic properties and contribute to the healing of skin wounds. Importantly, these beneficial effects remain consistent across various experimental animal models, methods of administration, exosome concentrations, number of administrations, and sources of exosomes.

Some meta-analyses have been published with positive outcomes that corroborate these findings. Qiao et al. and Masawa et al. demonstrated that exosome-based therapies improve angiogenesis, reepithelization, and collagen deposition while decreasing local inflammation. Therefore, the results indicate that the treatments accelerate wound healing [3,177].

## 5. Conclusions

Addressing the need for effective therapeutic options to promote skin regeneration remains a significant goal, as it poses an ongoing challenge to public health. This challenge is expected to intensify with the increasing population suffering from chronic diseases and the general aging of the population associated with an increase in average life expectancy. As a potential biological therapeutic approach, exosome-based therapies emerge as a promising strategy for wound healing.

This comprehensive systematic review highlights the great potential of exosomes as therapeutic options for non-healing chronic wounds. In summary, exosome treatment has shown consistent positive outcomes, including enhanced wound-closure rates, stimulation of local angiogenesis and reepithelization, and increased collagen deposition. Moreover, exosomes have demonstrated the ability to reduce scar formation, alleviate local inflammation, promote increased granulation tissue formation, and enhance the proliferation and migration of dermal fibroblasts. These findings underscore the therapeutic efficacy of exosomes in promoting wound healing. The field has also witnessed significant advancements in the last 5 years by combining exosomes with innovative engineering strategies. Exosome-based therapies have emerged as promising tools for wound healing, with advantages such as abundant sources; ease of preparation, storage, and transportation; as well as minimal immunogenicity.

Despite the potential of exosomes in wound treatment, challenges persist. The methodology associated with the use of exosomes-based therapies in wound healing remains highly heterogeneous. The mechanisms underlying exosome biogenesis, cellular interactions, and communication remain largely unknown. Comprehensive research is necessary to deepen our understanding of exosome biology for safe and effective applications. Standardized preparation methods and technical issues pose obstacles that impact the pharmacokinetic and pharmacodynamic performance of exosomes. Precise cargo loading, large-scale production, targeted delivery, reduced contamination, and cytotoxicity all require further exploration. Nonetheless, standardization and optimization of exosomes isolation and purification methods, scalability for clinical applications, and enhancing therapeutic potential through tissue engineering strategies are important challenges to address.

Further research endeavors are imperative to facilitate the commercial availability and clinical application of these treatments. 

The considerable variability in cell sources, types, biomaterials, and administration routes under investigation shows the urgent need of further research in this field. Moreover, the lack of comparative studies exploring different cell sources/types, administration routes, or even biomaterials is a critical gap that must be addressed. Furthermore, the predominant use of rodent-based animal models raises concerns, as limited progress has been made in advancing toward more complex in vivo models that closely resemble human physiology.

This study also has certain limitations, primarily due to the potential bias associated with study design and methodology of the included studies. To address these limitations, it is crucial that future research incorporates measures to mitigate bias, such as randomization, blinding, and standardized protocols. Due to the absence of a universally agreed-upon set of techniques for isolating, characterizing, and applying exosomes, we were unable to strictly enforce inclusion and exclusion criteria in our study. Additionally, the scope of our study did not allow for a comprehensive comparison between various wound-healing models. It is worth noting that a bias may have been introduced during data collection and reporting due to the prevalent practice of publishing only successful or efficacious studies.

To achieve a commercially viable and widely accessible range of therapeutic options, several key objectives must be pursued in the future. Standardizing methodologies is paramount to ensure more reliable and comparable results. In addition, the inclusion of more complex animal models that closely mimic the human species will enable the effective translation of research outcomes. These collective efforts will drive the field closer to its ultimate goal of achieving large-scale production and widespread availability of exosome-based therapeutic option for wound healing.

To overcome the limitations associated with solely relying on rodent models, we suggest expanding our investigation to include a diverse range of models, including non-rodent and non-animal alternatives. Furthermore, it is crucial to extend the follow-up period to thoroughly evaluate the impact of exosomes on the maturation and scarring of healed skin as well as the immunological response. Additionally, future research endeavors should encompass various skin lesions, such as burns, ulcers, and incisional ischemic lesions, and utilize more representative models of diabetic type 2 and non-healing wounds.

In conclusion, our research findings support the hypothesis that exosomes have great potential as therapeutic options for wound-healing applications. However, it is crucial to establish a consensus regarding the definition and standardization of variables related to exosome isolation, quantification, administration, and reporting. Multidisciplinary research is crucial to address the scientific questions and technical challenges associated with exosomes and to bridge the gap between experimental studies and commercialization. Continued advancements in the field will pave the way for the translation of exosome-based therapies into clinically applicable approaches. With ongoing advancements in the field of exosome research and continued efforts in addressing these challenges, exosome-based therapies hold great promise for future clinical applications. Achieving unanimous agreement on these variables will undoubtedly facilitate the advancement of the exosome field and pave the way for a promising future, allowing its translation into clinical practice.

## Figures and Tables

**Figure 1 biomedicines-11-02099-f001:**
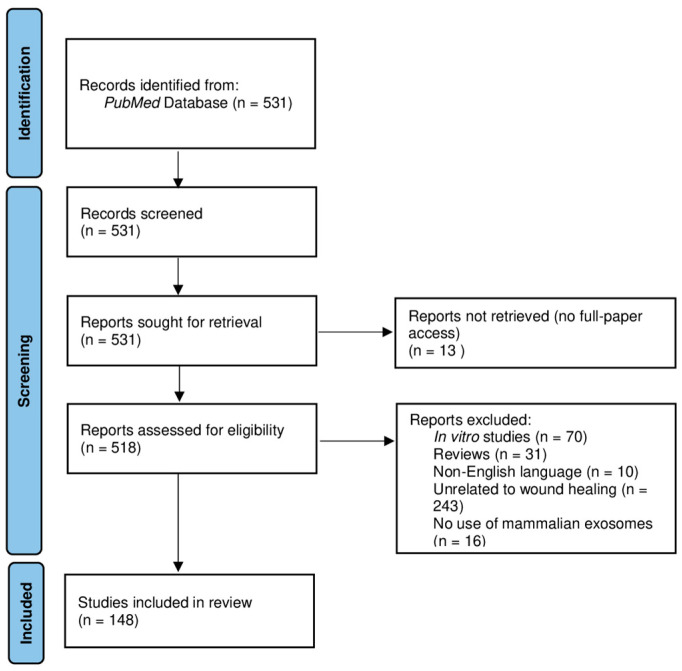
PRISMA Flow Diagram that summarizes the selection process.

**Figure 2 biomedicines-11-02099-f002:**
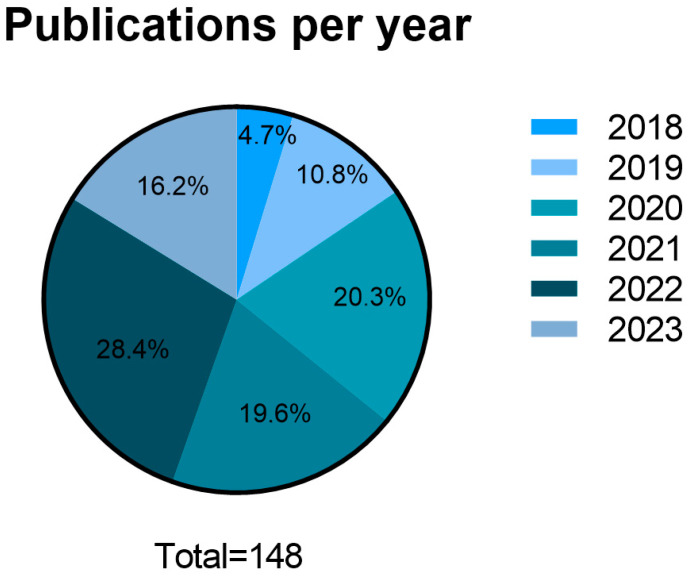
Graphical representation of publication distribution per year, from 2018 to June 2023.

**Figure 3 biomedicines-11-02099-f003:**
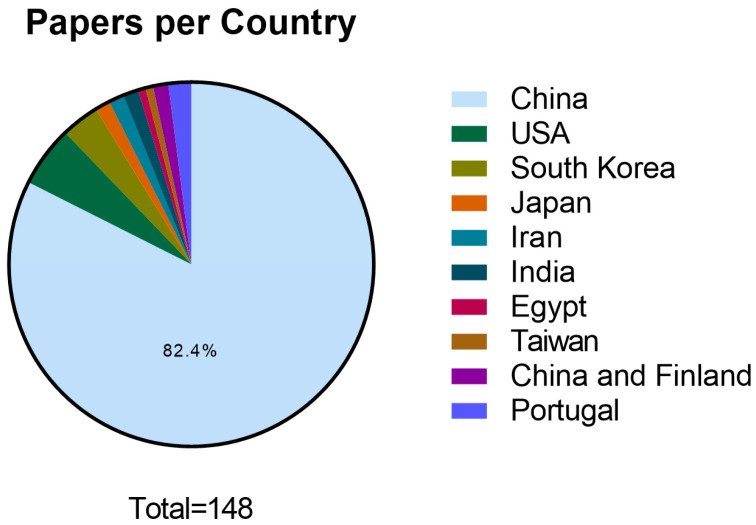
Graphical representation of publication distribution per corresponding authors country (2018–June 2023).

**Figure 4 biomedicines-11-02099-f004:**
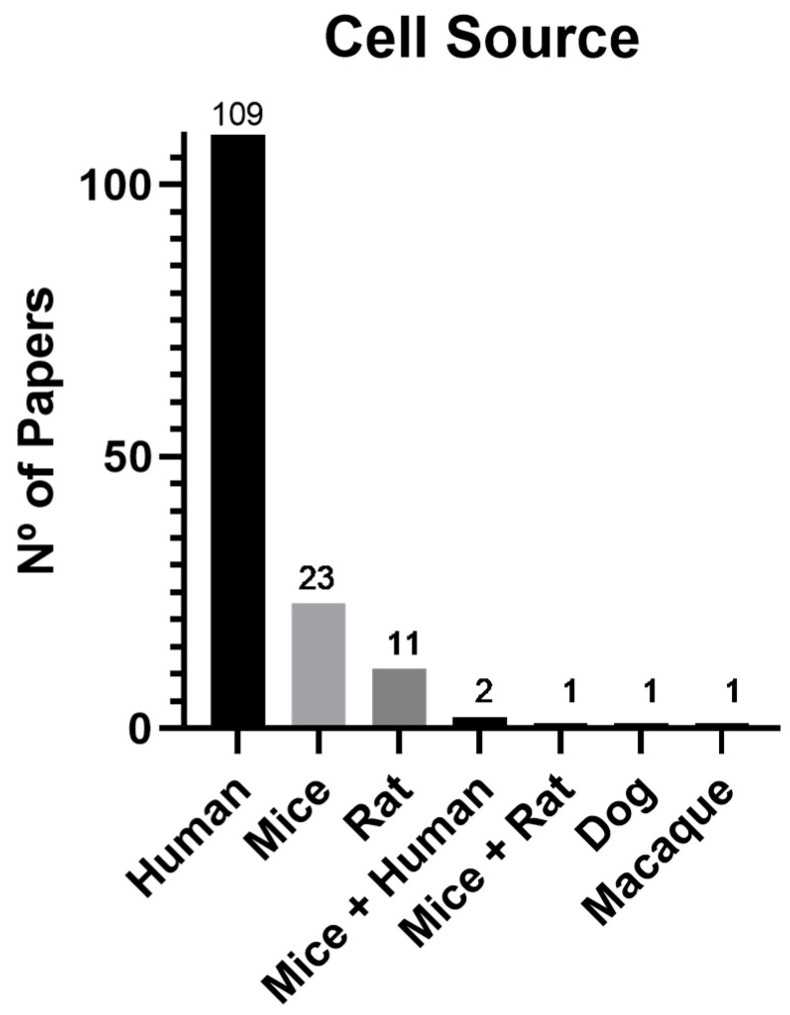
Graphical representation of cell tissue source distribution in the scientific literature between 2018 and June 2023.

**Figure 5 biomedicines-11-02099-f005:**
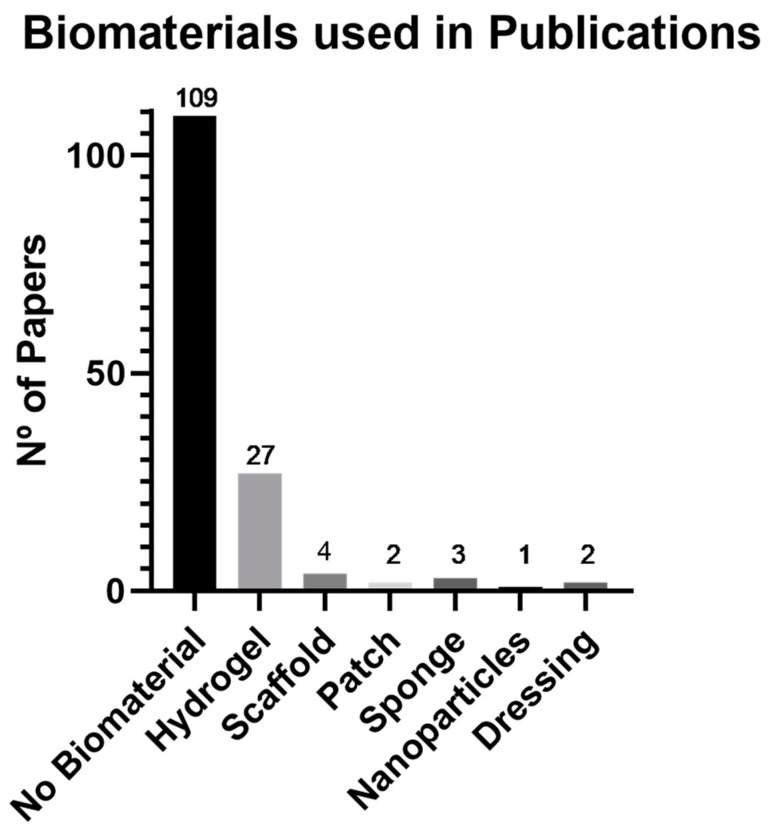
Graphical representation of the biomaterials used in publications related to wound healing between 2018 and June 2023.

**Figure 6 biomedicines-11-02099-f006:**
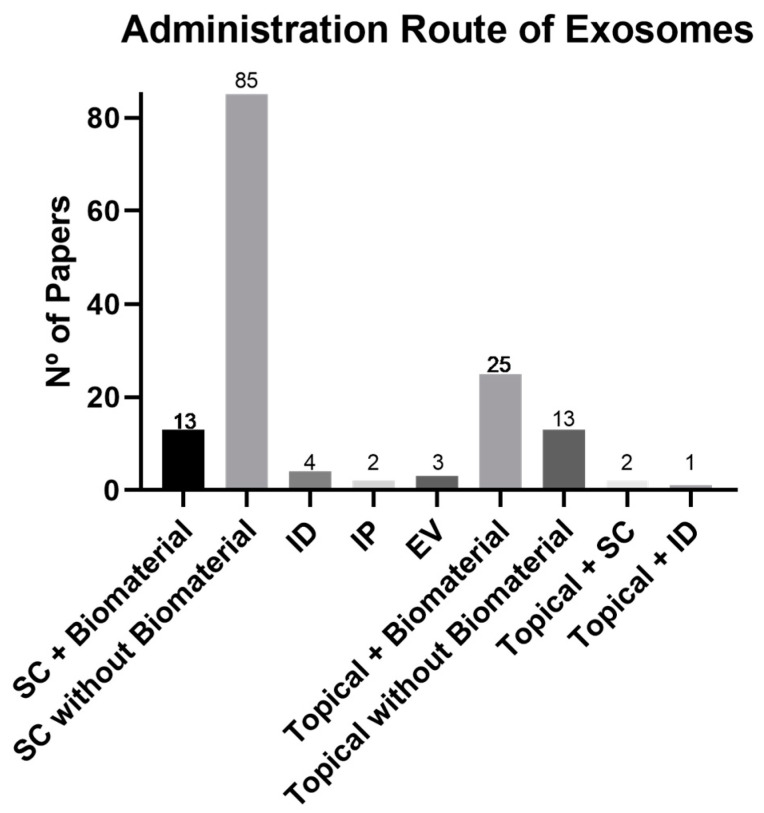
Graphical representation of the administration route of exosomes used in wound healing between 2018 and June 2023. EV, endovenous; ID, intradermal; IP, intraperitoneal; SC, subcutaneous.

**Figure 7 biomedicines-11-02099-f007:**
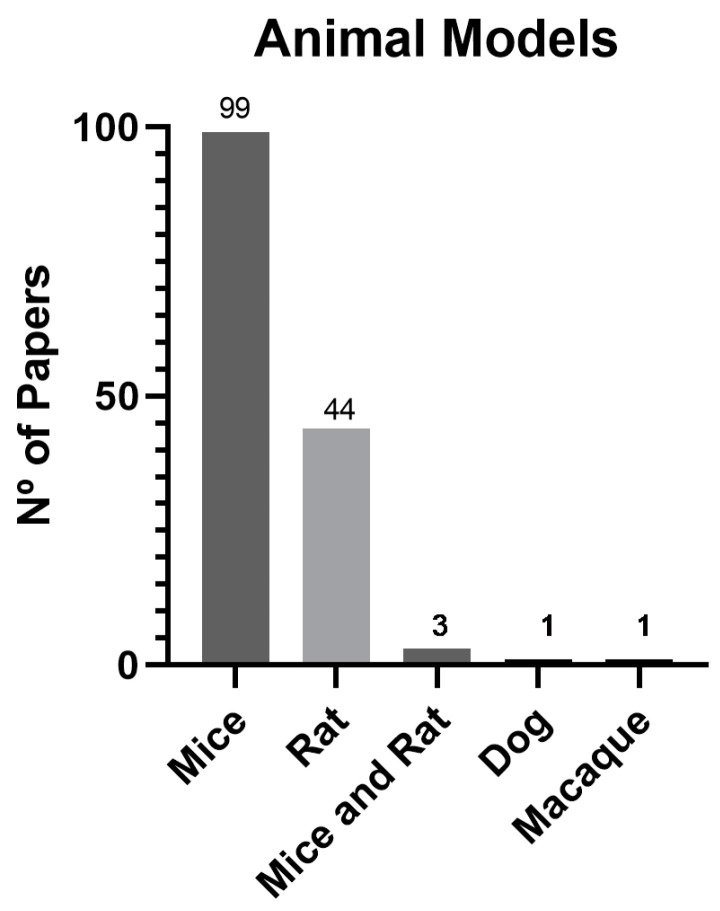
Graphical representation of the animal models used in wound healing between 2018 and June 2023.

**Table 1 biomedicines-11-02099-t001:** Summary of the retrieved data from the 148 papers. x—No biomaterial.

Ref	Year	Country	Cell Source	Cell Type	Biomaterial	Administration Route	Animal Models
[12]	2019	China	Human	ADSC	Hydrogel	SC Injection	Mice
[13]	2020	China	Human	UCMSC	Hydrogel	SC Injection	Rat
[14]	2018	China	Human	UCMSC	x	SC Injection	Mice
[15]	2020	China	Human	BMSC	x	SC Injection	Rat
[16]	2022	China	Human	UVEC	Patch	Patch	Mice
[17]	2022	China	Human	ADSC	Hydrogel	Topical	Mice
[18]	2020	China	Human	BMSC	x	SC Injection	Rat
[19]	2021	China	Rat	BMSC	x	SC Injection	Rat
[20]	2021	China	Human	ADSC	x	SC Injection	Mice
[21]	2018	China	Human	ADSC	x	SC Injection	Rat
[22]	2022	China	Mice	BMSC	Hydrogel	SC Injection	Mice
[23]	2022	USA	Human	Epidermal	x	SC Injection	Mice
[24]	2020	USA	Mice	Keratinocyte	x	SC Injection	Mice
[25]	2022	China	Human	ADSC	x	SC Injection	Mice
[26]	2020	China	Human	BMSC	x	SC Injection	Rat
[27]	2022	China	Rat	BMSC	Hydrogel	Topical	Rat
[28]	2020	China	Human	Peripheral Blood	x	SC Injection	Mice
[29]	2020	India	Rat	ADSC	Scaffold	Scaffold	Rat
[30]	2022	China	Human	DPs	x	SC Injection	Mice
[31]	2019	China	Human	ADSC	Scaffold	Scaffold	Mice
[32]	2021	China	Human	UCMSC	x	EV Injection	Rat
[33]	2019	China	Human	BMSC	x	SC Injection	Mice
[34]	2022	Republic of Korea	Mice	BMSC	Hydrogel	SC Injection	Mice
[35]	2020	China	Human	ADSC	x	SC Injection	Rat
[36]	2020	China	Mice	BMSC	x	ID Injection	Mice
[37]	2021	China	Human	ADSC	x	SC Injection	Mice
[38]	2022	China	Human	Epidermal	Hydrogel	SC Injection	Mice
[39]	2022	Republic of Korea	Human	ADSC	x	SC Injection	Mice
[40]	2022	China	Human	ADSC	x	SC Injection	Mice
[41]	2019	Republic of Korea	Mice	BMSC	x	SC Injection	Mice
[42]	2020	China	Human	BMSC	x	SC Injection	Mice
[43]	2023	China	Mice	ADSC	Hydrogel	SC Injection	Mice
[44]	2021	China	Human	ADSC	Scaffold	Scaffold	Mice
[45]	2021	China	Mice	Serum	x	SC Injection	Mice
[46]	2022	China	Mice	Fibroblast	x	ID Injection	Mice
[47]	2018	China	Human	ADSC	x	SC and ID Injection	Mice
[48]	2019	China	Human	ADSC	x	SC Injection	Mice
[49]	2023	China	Human	Placenta	Patch	Patch	Mice
[50]	2019	China	Human	Embryonic	x	Topical	Mice
[51]	2022	China	Human	UCMSC	x	SC Injection	Mice
[52]	2022	China	Mice	ADSC	x	SC Injection	Mice
[53]	2021	China	Rat and Mice	Serum	x	SC Injection	Mice
[54]	2019	China	Human	Macrophage	x	SC Injection	Mice
[55]	2022	China	Human	UCMSC	x	Topical	Mice
[56]	2019	China	Macaque	iPSCs	x	Topical	Macaque
[57]	2022	China	Mice	ADSC	x	SC Injection	Mice
[58]	2022	China	Human	UCMSC	x	SC Injection	Mice
[59]	2020	China	Human	ADSC	x	SC Injection	Mice
[60]	2022	China	Human	UCMSC	x	SC Injection	Rat
[61]	2020	China	Human	UCMSC	x	SC Injection	Mice
[62]	2020	China	Human	UVEC	Hydrogel	Topical	Rat
[63]	2022	China	Human	iPSCs	Hydrogel	Topical	Mice
[64]	2023	China	Mice	ADSC	x	SC Injection	Mice
[65]	2020	China	Human	Amniotic Membrane	x	SC Injection	Mice
[66]	2022	China	Human	UVEC	x	SC Injection	Mice
[67]	2021	China	Human	ADSC	x	Injection and Topical	Mice
[68]	2021	China	Human	UCMSC	Hydrogel	Topical	Mice
[69]	2019	China	Human	UVEC	x	SC Injection	Rat
[70]	2023	China	Human	UVEC	Hydrogel	Microneedle	Rat
[71]	2021	China	Human	UCMSC	Hydrogel	SC Injection	Rat
[72]	2020	China	Human	UVEC	Hydrogel	SC Injection	Rat
[73]	2022	China	Rat	BMSC	x	SC Injection	Rat
[74]	2022	China	Human	ADSC	x	SC Injection	Rat
[75]	2023	China	Human	UCMSC	x	SC Injection	Mice
[76]	2022	China	Mice	ADSC	Hydrogel	Topical	Rat
[77]	2022	China	Human	ADSC	x	SC Injection	Mice
[78]	2019	Japan	Human	Epithelial	x	Topical	Rat
[79]	2022	China	Human	DPs	x	SC Injection	Mice
[80]	2021	China	Rat	Dermal	x	SC Injection	Rat
[81]	2022	Portugal	Human	UCMSC	x	SC Injection	Rat
[82]	2022	China	Rat	Placenta and ADSC	x	SC Injection	Rat
[83]	2022	China	Human	UCMSC	Hydrogel	SC Injection	Mice
[84]	2021	China	Human	ADSC	x	SC Injection	Rat
[85]	2022	China	Human	UVEC	x	SC Injection	Mice
[86]	2019	China	Human	Fetal dermal	x	SC Injection	Mice
[87]	2020	China	Mice	BMSC	x	Topical	Mice
[88]	2023	Iran	Human	Fetal dermal	x	Topical	Rat
[89]	2022	Taiwan	Mice	ADSC and dermal	x	Topical	Mice
[90]	2020	China and Finland	Human	ADSC	x	IP Injection	Mice
[91]	2022	USA	Human	BMSC	x	SC Injection	Mice and Rat
[92]	2020	China	Human	BMSC	x	SC Injection	Mice
[93]	2021	China	Human	UCMSC	x	SC Injection	Rat
[94]	2022	China	Human	ADSC	x	SC Injection	Mice
[95]	2020	India	Rat	ADSC	Scaffold	Scaffold	Rat
[96]	2019	China	Human	Placenta	Hydrogel	SC Injection	Mice
[97]	2020	China	Human	BMSC	x	SC Injection	Rat
[98]	2023	China	Rat	BMSC and plasma	Hydrogel	Topical	Rat
[99]	2021	China and Finland	Human	ADSC	x	IP Injection	Mice
[100]	2021	China	Human	UCMSC	x	SC Injection	Mice
[101]	2020	China	Human	Epidermal	x	SC Injection	Rat
[102]	2020	China	Human	ADSC	Hydrogel	Topical	Rat
[103]	2018	China	Human	Plasma	Sponge	Sponge	Rat
[104]	2021	China	Human	UCMSC and ADSC	x	SC Injection	Mice
[105]	2019	Iran	Human	Menstrual Blood	x	ID Injection	Mice
[106]	2020	China	Human	Saliva	x	SC Injection	Mice
[107]	2023	China	Human	ADSC	Hydrogel	SC Injection	Mice
[108]	2021	China	Human	Peripheral Blood	x	SC Injection	Mice
[109]	2021	China	Mice	Plasma	x	SC Injection	Mice
[110]	2018	Japan	Human	iPSCs	x	SC Injection	Mice
[111]	2022	USA	Mice and Human	Plasma	x	Topical	Mice
[112]	2023	China	Human	UCMSC	x	Topical	Mice
[113]	2020	China	Human	Amnion	x	SC Injection	Rat
[114]	2022	China	Human	Keratinocyte	x	SC Injection	Mice
[115]	2022	China	Mice	ADSC	Hydrogel	SC Injection	Rat
[116]	2021	China	Mice	Dermal	x	ID Injection	Mice
[117]	2022	USA	Mice	Skin	Sponge	Sponge	Mice
[118]	2023	China	Human	UCMSC	x	SC Injection	Mice
[119]	2019	China	Human	BMSC	x	SC Injection	Rat
[120]	2021	China	Human	Amniotic Fluid	x	SC Injection	Rat
[121]	2022	China	Human	ADSC	x	SC Injection	Mice
[122]	2018	China	Human	Amniotic Fluid	x	SC Injection	Mice
[123]	2021	China	Human	ADSC	x	SC Injection	Rat
[124]	2020	China	Human	Peripheral Blood	x	Injection	Mice
[125]	2022	China	Rat	ADSC	Hydrogel	Topical	Rat
[126]	2020	China	Human	UCMSC	Nanoparticles	EV Injection	Rat
[127]	2023	Republic of Korea	Human	UCMSC	x	Injection and Topical	Mice and Rat
[128]	2022	China	Human	Gingival	x	SC Injection	Mice
[129]	2021	China	Human	UCMSC	Hydrogel	Topical	Rat
[130]	2022	China	Human	BMSC	x	EV Injection	Mice
[131]	2023	China	Human	Keratinocyte	x	SC Injection	Mice
[132]	2021	China	Human	DPs	x	Topical	Mice
[133]	2020	China	Human	UCMSC	Dressing	Topical	Mice
[134]	2023	China	Human	UCMSC	x	SC Injection	Mice
[135]	2021	China	Human	Embryonic	x	SC Injection	Rat
[136]	2022	China	Mice	Dendritic epidermal T cells	x	SC Injection	Mice
[137]	2021	USA	Mice	Skin	Sponge	Topical	Mice
[138]	2023	China	Human	Epidermal	x	SC Injection	Mice
[139]	2022	China	Human	UVEC	x	SC Injection	Mice
[140]	2019	Portugal	Human	UCMSC	Hydrogel	Topical	Mice
[141]	2023	China	Mice	Macrophage	x	Topical	Rat
[142]	2021	Portugal	Human	UCMSC	Hydrogel	Topical	Mice
[143]	2023	China	Human	UCMSC	x	SC Injection	Mice
[144]	2021	China	Human	BMSC	x	SC Injection	Mice
[145]	2018	USA	Mice and Human	BMSCs, Skin and Gingiva	x	SC Injection	Mice
[146]	2023	China	Mice	ADSC	x	SC Injection	Mice
[147]	2023	China	Human	Plasma	x	SC Injection	Mice
[148]	2023	China	Human	Hair follicle	x	SC Injection	Mice
[149]	2020	China	Human	ADSC	x	SC Injection	Mice
[150]	2023	China	Rat	ADSC	x	SC Injection	Rat
[151]	2023	China	Mice	BMSC	Dressing	Topical	Rat
[152]	2020	China	Human	Peripheral Blood	x	SC Injection	Mice
[153]	2023	China	Rat	BMSC	Hydrogel	SC Injection	Mice and Rat
[154]	2019	Republic of Korea	Human	Fibroblast	x	Topical	Mice
[155]	2020	China	Human	ADSC	x	SC Injection	Mice
[156]	2023	USA	Human	Placenta	x	Topical	Mice
[157]	2023	China	Human	ADSC	x	SC Injection	Mice
[158]	2021	Egypt	Dog	BMSC	Hydrogel	Topical	Dog
[159]	2021	China	Human	ADSC	x	SC Injection	Mice

**Table 2 biomedicines-11-02099-t002:** Summary of the most used cell types of the most common species. X—Not applicable.

	Human (109 Studies)	Mice (23 Studies)	Rat (11 Studies)
BMSC	11 (10.1%)	6 (26.1%)	4 (36.4%)
ADSC	29 (26.6%)	7 (30.4%)	4 (36.4%)
UCMSC	25 (22.9%)	X	X
UVECS	8 (7.3%)	X	X
DP	3 (2.8%)	X	X
Epidermal	4 (3.7%)	X	X
Peripherical blood	4 (3.7%)	X	X
Placenta	3 (2.8%)	X	X
Others	22 (20.2%)	10 (43.5%)	3 (27.3%)

## Data Availability

Further data on the reported results are available from the corresponding author on request.

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
