# Peer review of "Advancements and Insights in Exosome-Based Therapies for Wound Healing: A Comprehensive Systematic Review (2018–June 2023)"

_biomedicines, 2023, doi:10.3390/biomedicines11082099_

Round 1

Reviewer 1 Report

I have carefully reviewed the manuscript titled "Methodological Heterogeneity and Animal Model Suitability in Exosome-based Therapies for Wound Healing" submitted to Biomedicines. I appreciate the authors' effort in addressing an important topic in the field of exosome-based therapies for wound healing. However, after thorough evaluation, I cannot recommend this manuscript for publication in its current form due to several significant concerns.

While the abstract provides an overview of the selected papers and highlights the heterogeneity in methodological approaches, it lacks a comprehensive analysis of the role of exosomes in wound healing. As a review article, it is expected to provide scientific insights and draw meaningful conclusions based on the available research data. Therefore, the authors should expand the discussion to include an analysis of the reported effects of exosomes on wound healing, supported by scientific evidence from the selected papers. This will enhance the significance and contribution of the manuscript to the field. 

Additionally, the conclusion should be revised to provide a more definitive and focused summary of the key findings and implications of the reviewed studies. It is important for a review article to offer valuable insights and recommendations for future research or clinical applications. The authors should consider discussing the challenges and opportunities in translating exosome-based therapies into clinically applicable approaches, and propose potential strategies for bridging the gap between experimental studies and commercialization. 

Minor editing of English language required

Author Response

Answer to reviewer 1

Dear reviewer 1:

Thank you very much for the feedback on this review phase, and also for the suggestions made, which received the best attention from us. The authors inform that the final document has been revised and all suggestions made by the reviewers have been introduced, being duly identified in the final document with highlight. This review also made it possible to identify and correct some errors and typos as well as improve the general level of English.

The changes made to the document are described below. All changes introduced and highlighted text segments appear in the final document highlighted in yellow.

  1. While the abstract provides an overview of the selected papers and highlights the heterogeneity in methodological approaches, it lacks a comprehensive analysis of the role of exosomes in wound healing.

The abstract and introduction have been improved in order to add a comprehensive analysis of the role of exosomes in wound healing. The new sections are transcribed below:

Abstract:

“To assess the current state of research in this field, a systematic review was performed involving studies conducted and published over the past five years.”

“The findings revealed that exosome-based treatments improve wound healing by increasing angiogenesis, reepithelization, collagen deposition and decreasing scar formation. Furthermore, there was significant variability in terms of cell sources and types, biomaterials, and administration routes under investigation, indicating the need for further research in this field.”

Introduction:

“Previous research has highlighted the crucial role of exosomes in facilitating cell-to-cell communication, namely by sharing their cargo as miRNAs and proteins. Several studies have indicated that exosomes obtained from stem cells have the potential to assist and promote tissue repair. This is attributed to their unique advantages, including exceptional stability, minimal risk of immune rejection, targeted delivery to specific tissues, straightforward control of dosage and definable concentration [8].”

“The quality of wound healing relies on the fibroblast’s migration and proliferation, as well as collagen synthesis and deposition. Exosomes, with their abundant content of RNAs and proteins relevant to fibroblast functions, are thought to facilitate these processes. This optimization of fibroblast activities ultimately contributes to the accelerated wound healing mechanism [8].

Multiple studies have demonstrated the therapeutic potential of exosomes in various stages of wound healing. In the inflammation phase, exosomes have been shown to modulate immune cells and resident tissue cells, leading to a reduction in uncontrolled inflammatory responses. During the proliferation phase, exosomes play a role in wound closure by activating endothelial cells and fibroblasts. This activation promotes a proangiogenic environment and initiates the deposition of extracellular matrix. In the remodeling phase, exosomes influence the balance between matrix metalloproteinases and tissue inhibitors of matrix metalloproteinases, favoring optimal wound healing outcomes. Exosome therapy also enhances wound healing by stabilizing and stimulating a wide range of mediators involved in each phase [10].”

“The aim of this systematic review was to access the exosome-based therapies wound healing effects…”

  1. As a review article, it is expected to provide scientific insights and draw meaningful conclusions based on the available research data. Therefore, the authors should expand the discussion to include an analysis of the reported effects of exosomes on wound healing, supported by scientific evidence from the selected papers. This will enhance the significance and contribution of the manuscript to the field. 

The authors have added the following sentences in the discussion, in which the reported effects of exosomes in wound healing are described:

“All the selected papers consistently reported positive outcomes in vivo, highlighting the ability of exosomes to promote wound healing.

Exosome treatment was found to enhance wound closure rates, stimulate local angiogenesis and reepithelization, and facilitate collagen deposition [12, 14, 17, 18]. Furthermore, exosomes promoted a reduction in scar formation and decreased local inflammation in multiple studies [18, 23, 158]. Additionally, exosome treatments resulted in increased granulation tissue formation and enhanced proliferation and migration of dermal fibroblasts [65, 155, 157].

The findings consistently indicate that exosomes possess therapeutic properties and contribute to the healing of skin wounds. Importantly, these beneficial effects remain consistent across various experimental animal models, methods of administration, exosome concentrations, number of administrations, and sources of exosomes.

Some meta-analyses have been published with positive outcomes that corroborate these findings. Qiao et al and Masawa et al demonstrated that exosome-based therapies improve angiogenesis, reepithelization and collagen deposition, while decreasing local inflammation. Therefore, the results indicate that the treatments accelerate wound healing [3, 177].”

  1. Additionally, the conclusion should be revised to provide a more definitive and focused summary of the key findings and implications of the reviewed studies. It is important for a review article to offer valuable insights and recommendations for future research or clinical applications. The authors should consider discussing the challenges and opportunities in translating exosome-based therapies into clinically applicable approaches, and propose potential strategies for bridging the gap between experimental studies and commercialization. 

The authors have added several sentences to the conclusion, in order to incorporate the reviewer’s suggestions:

“Addressing the need for effective therapeutic options to promote skin regeneration remains a significant goal, as it poses an ongoing challenge to public health. This challenge is expected to intensify with the increasing population suffering from chronic diseases and the general aging of the population associated with an increase in average life expectancy. As a potential biological therapeutic approach, exosome-based therapies emerge as a promising strategy for wound healing.”

“In summary, exosome treatment has shown consistent positive outcomes, including enhanced wound closure rates, stimulation of local angiogenesis and reepithelization, and increased collagen deposition. Moreover, exosomes have demonstrated the ability to reduce scar formation, alleviate local inflammation, promote increased granulation tissue formation, and enhance the proliferation and migration of dermal fibroblasts. These findings underscore the therapeutic efficacy of exosomes in promoting wound healing. The field has also witnessed significant advancements in the last 5 years, combining exosomes with innovative engineering strategies. Exosome-based therapies have emerged as promising tools for wound healing, with advantages such as abundant sources, ease of preparation, storage, and transportation, as well as minimal immunogenicity.

Despite the potential of exosomes in wound treatment, challenges persist.”

“The mechanisms underlying exosome biogenesis, cellular interactions, and communication remain largely unknown. Comprehensive research is necessary to deepen our understanding of exosome biology for safe and effective applications. Standardized preparation methods and technical issues pose obstacles that impact the pharmacokinetic and pharmacodynamic performance of exosomes. Precise cargo loading, large-scale production, targeted delivery, reduced contamination, and cytotoxicity all require further exploration. Nonetheless, standardization and optimization of exosomes isolation and purification methods, scalability for clinical applications, and enhancing therapeutic potential through tissue engineering strategies are important challenges to address.”

“Due to the absence of a universally agreed-upon set of techniques for isolating, characterizing, and applying exosomes, we were unable to strictly enforce inclusion and exclusion criteria in our study. Additionally, the scope of our study did not allow for a comprehensive comparison between various wound healing models. It is worth noting that a bias may have been introduced during data collection and reporting due to the prevalent practice of publishing only successful or efficacious studies.”

“To overcome the limitations associated with solely relying on rodent models, we suggest expanding our investigation to include a diverse range of models, including non-rodent and non-animal alternatives. Furthermore, it is crucial to extend the follow-up period to thoroughly evaluate the impact of exosomes on the maturation and scarring of healed skin, as well as the immunological response. Additionally, future research endeavors should encompass various skin lesions such as burns, ulcers, incisional ischemic lesions, and utilize more representative models of diabetic type 2 and non-healing wounds.

In conclusion, our research findings support the hypothesis that exosomes, have great potential as therapeutic options for wound healing applications. However, it is crucial to establish a consensus regarding the definition and standardization of variables related to exosome isolation, quantification, administration, and reporting. Multidisciplinary research is crucial to address the scientific questions and technical challenges associated with exosomes and to bridge the gap between experimental studies and commercialization. Continued advancements in the field will pave the way for the translation of exosome-based therapies into clinically applicable approaches. With ongoing advancements in the field of exosome research and continued efforts in addressing these challenges, exosome-based therapies hold great promise for future clinical applications. Achieving unanimous agreement on these variables will undoubtedly facilitate the advancement of the exosome field and pave the way for a promising future, allowing its translation into clinical practice.”

Reviewer 2 Report

This is an interesting review about the recent development in wound healing based on Exosome-based therapies.

The topic of the paper well fits with the scope of the journal and is of interest for scientists working in a multi-disciplinary context.

It is an opinion of this reviewer that the paper should be eventually published, although some key revisions should be performed in the last section of the paper. In the conclusion, indeed, authors should strengthen they critical analysis of the data and point out the key perspective in the field. This point should be carefully addressed by the authors before further processing of the paper.

Minor point: problems with some images but maybe related to the pdf conversion

Author Response

Answer to reviewer 2

Dear reviewer 2:

Thank you very much for the feedback on this review phase, and also for the suggestions made, which received the best attention from us. The authors inform that the final document has been revised and all suggestions made by the reviewers have been introduced, being duly identified in the final document with highlight. This review also made it possible to identify and correct some errors and typos as well as improve the general level of English.

The changes made to the document are described below. All changes introduced and highlighted text segments appear in the final document highlighted in yellow.

  1. In the conclusion, indeed, authors should strengthen they critical analysis of the data and point out the key perspective in the field.

The following sentences were added to the conclusion, in order to incorporate the reviewer’s suggestions:

“Addressing the need for effective therapeutic options to promote skin regeneration remains a significant goal, as it poses an ongoing challenge to public health. This challenge is expected to intensify with the increasing population suffering from chronic diseases and the general aging of the population associated with an increase in average life expectancy. As a potential biological therapeutic approach, exosome-based therapies emerge as a promising strategy for wound healing.”

“In summary, exosome treatment has shown consistent positive outcomes, including enhanced wound closure rates, stimulation of local angiogenesis and reepithelization, and increased collagen deposition. Moreover, exosomes have demonstrated the ability to reduce scar formation, alleviate local inflammation, promote increased granulation tissue formation, and enhance the proliferation and migration of dermal fibroblasts. These findings underscore the therapeutic efficacy of exosomes in promoting wound healing. The field has also witnessed significant advancements in the last 5 years, combining exosomes with innovative engineering strategies. Exosome-based therapies have emerged as promising tools for wound healing, with advantages such as abundant sources, ease of preparation, storage, and transportation, as well as minimal immunogenicity.

Despite the potential of exosomes in wound treatment, challenges persist.”

“The mechanisms underlying exosome biogenesis, cellular interactions, and communication remain largely unknown. Comprehensive research is necessary to deepen our understanding of exosome biology for safe and effective applications. Standardized preparation methods and technical issues pose obstacles that impact the pharmacokinetic and pharmacodynamic performance of exosomes. Precise cargo loading, large-scale production, targeted delivery, reduced contamination, and cytotoxicity all require further exploration. Nonetheless, standardization and optimization of exosomes isolation and purification methods, scalability for clinical applications, and enhancing therapeutic potential through tissue engineering strategies are important challenges to address.”

“Due to the absence of a universally agreed-upon set of techniques for isolating, characterizing, and applying exosomes, we were unable to strictly enforce inclusion and exclusion criteria in our study. Additionally, the scope of our study did not allow for a comprehensive comparison between various wound healing models. It is worth noting that a bias may have been introduced during data collection and reporting due to the prevalent practice of publishing only successful or efficacious studies.”

“To overcome the limitations associated with solely relying on rodent models, we suggest expanding our investigation to include a diverse range of models, including non-rodent and non-animal alternatives. Furthermore, it is crucial to extend the follow-up period to thoroughly evaluate the impact of exosomes on the maturation and scarring of healed skin, as well as the immunological response. Additionally, future research endeavors should encompass various skin lesions such as burns, ulcers, incisional ischemic lesions, and utilize more representative models of diabetic type 2 and non-healing wounds.

In conclusion, our research findings support the hypothesis that exosomes, have great potential as therapeutic options for wound healing applications. However, it is crucial to establish a consensus regarding the definition and standardization of variables related to exosome isolation, quantification, administration, and reporting. Multidisciplinary research is crucial to address the scientific questions and technical challenges associated with exosomes and to bridge the gap between experimental studies and commercialization. Continued advancements in the field will pave the way for the translation of exosome-based therapies into clinically applicable approaches. With ongoing advancements in the field of exosome research and continued efforts in addressing these challenges, exosome-based therapies hold great promise for future clinical applications. Achieving unanimous agreement on these variables will undoubtedly facilitate the advancement of the exosome field and pave the way for a promising future, allowing its translation into clinical practice.”

Round 2

Reviewer 1 Report

Thank you for sharing the revised version. 

Having reviewed the revised manuscript, I'm pleased to see the authors have adequately addressed my previous comments. In my view, the manuscript now meets the necessary standard for acceptance and publication.

Reviewer 2 Report

Authors addressed all criticisms. This reviewer is recommending publication of the paper in its current form